# Association of genetic liability to smoking initiation with e-cigarette use in young adults: A cohort study

Jasmine N. Khouja[1,2,3]*, Robyn E. Wootton[1,2], Amy E. Taylor[2,4], George Davey Smith[1,2], Marcus R. Munafò[1,3,4]

1 MRC Integrative Epidemiology Unit at the University of Bristol, Bristol, United Kingdom, 2 Bristol Medical School: Population Health Sciences, University of Bristol, Bristol, United Kingdom, 3 School of Psychological Science, University of Bristol, Bristol, United Kingdom, 4 NIHR Biomedical Research Centre at the University Hospitals Bristol NHS Foundation Trust and the University of Bristol, Bristol, United Kingdom

* jasmine.khouja@bristol.ac.uk

**Data Availability Statement:** The data used in this study are available on request to the ALSPAC Executive (alspac-exec@bristol.ac.uk). The ALSPAC data management plan describes in detail

## Abstract

### Background

Tobacco smoking and e-cigarette use are strongly associated, but it is currently unclear whether this association is causal, or due to shared factors that influence both behaviours such as a shared genetic liability. The aim of this study was to investigate whether polygenic risk scores (PRS) for smoking initiation are associated with ever use of e-cigarettes.

### Methods and findings

Smoking initiation PRS were calculated for young adults ($N = 7,859$, mean age = 24 years, 51% male) of European ancestry in the Avon Longitudinal Study of Parents and Children, a prospective birth cohort study initiated in 1991. PRS were calculated using the GWAS & Sequencing Consortium of Alcohol and Nicotine use (GSCAN) summary statistics. Five thresholds ranging from $5 \times 10^{-8}$ to 0.5 were used to calculate 5 PRS for each individual. Using logistic regression, we investigated the association between smoking initiation PRS and the main outcome, self-reported e-cigarette use ($n = 2,894$, measured between 2016 and 2017), as well as self-reported smoking initiation and 8 negative control outcomes (socioeconomic position at birth, externalising disorders in childhood, and risk-taking in young adulthood). A total of 878 young adults (30%) had ever used e-cigarettes at 24 years, and 150 (5%) were regular e-cigarette users at 24 years. We observed positive associations of similar magnitude between smoking initiation PRS (created using the $p < 5 \times 10^{-8}$ threshold) and both smoking initiation (odds ratio (OR) = 1.29, 95% CI 1.19 to 1.39, $p < 0.001$) and ever e-cigarette use (OR = 1.24, 95% CI 1.14 to 1.34, $p < 0.001$) by the age of 24 years, indicating that a genetic predisposition to smoking initiation is associated with an increased risk of using e-cigarettes. At lower $p$-value thresholds, we observed an association between smoking initiation PRS and ever e-cigarette use among never smokers. We also found evidence of associations between smoking initiation PRS and some negative control outcomes, particularly when less stringent $p$-value thresholds were used to create the PRS, but

the policy regarding data sharing. Full instructions for applying for data access can be found here: http://www.bristol.ac.uk/alspac/researchers/access/. The ALSPAC study website contains details of all the data that are available (http://www.bristol.ac.uk/alspac/researchers/our-data/).

**Funding:** All authors are members of the Integrative Epidemiology Unit at the University of Bristol which is funded by the UK Medical Research Council (https://mrc.ukri.org/; grant numbers MC_UU_0001/1&7). The UK Medical Research Council and Wellcome (grant number 102215/2/13/2) and the University of Bristol (http://www.bristol.ac.uk/) provide core support for the Avon Longitudinal Study of Parents and Children (ALSPAC). A comprehensive list of grants funding is available on the ALSPAC website (http://www.bristol.ac.uk/alspac/external/documents/grant-acknowledgements.pdf). This research was specifically funded by a Cancer Research UK (https://www.cancerresearchuk.org/) grant to AET (grant number C54841/A20491). This publication is the work of the authors (JNK, REW, AET, GDS and MRM) who will serve as guarantors for the contents of this paper. AET and MRM are also supported by the NIHR Bristol Biomedical Research Centre (https://www.bristolbrc.nihr.ac.uk/) at University Hospitals Bristol NHS Foundation Trust and the University of Bristol. The sponsors played no role in the study design, data collection, data analysis, decision to publish, or preparation of the manuscript.

**Competing interests:** I have read the journal's policy and the authors of this manuscript have the following competing interests: GDS is a member of the Editorial Board of PLOS Medicine.

**Abbreviations:** ADHD, attention-deficit/hyperactivity disorder; ALSPAC, Avon Longitudinal Study of Parents and Children; CD, conduct disorder; e-cigarette, electronic cigarette; GSCAN, GWAS & Sequencing Consortium of Alcohol and Nicotine use; GWAS, genome-wide association studies; InSIDE, Instrument Strength Independent of Direct Effect; MAF, Minor Allele Frequency; MR, mendelian randomization; ODD, oppositional defiant disorder; OR, odds ratio; PRS, polygenic risk scores; SD, standard deviation; SEP, socioeconomic position; SNP, single nucleotide polymorphism; STROBE, Strengthening the Reporting of Observational Studies in Epidemiology.

also at the strictest threshold (e.g., gambling, number of sexual partners, conduct disorder at 7 years, and parental socioeconomic position at birth). However, this study is limited by the relatively small sample size and potential for collider bias.

## Conclusions

Our results indicate that there may be a shared genetic aetiology between smoking and e-cigarette use, and also with socioeconomic position, externalising disorders in childhood, and risky behaviour more generally. This indicates that there may be a common genetic vulnerability to both smoking and e-cigarette use, which may reflect a broad risk-taking phenotype.

## Author summary

### Why was this study done?

- Some individuals are more likely to smoke due to their genetics, but little is currently known about the genetic influences on e-cigarette use.

- Given that many people who use e-cigarettes have smoked before, it is likely that there may be an overlap between genetic influences on smoking and e-cigarette use.

- Such an overlap may explain why people who use e-cigarettes but have not smoked before are more likely to go on to start smoking later.

### What did the researchers do and find?

- We examined the association between genetic variants associated with smoking initiation and both e-cigarette use and risk-taking behaviour in a cohort of 2,894 young adults.

- Our results indicate that the genetic factors that influence smoking initiation are similarly related to e-cigarette use and risk-taking behaviours.

### What do these findings mean?

- Smoking may cause people to use e-cigarettes (i.e., to stop smoking), but there may also be an underlying genetic predisposition to risk-taking which influences the likelihood that someone will both smoke and use e-cigarettes.

- The findings could have important implications for policy—if young people are predisposed to both smoking and using e-cigarettes, bans which aim to prevent e-cigarette use may encourage smoking where only cigarettes are available.

## Introduction

There are an estimated 3.6 million electronic cigarette (e-cigarette) users in Great Britain [1], and evidence is growing that e-cigarettes are effective in helping tobacco smokers quit [2,3]. The use of e-cigarettes for smoking cessation is common among young adults in the United Kingdom [4]; therefore, it would be logical to assume that smoking causally influences e-cigarette use in this population. However, some studies have shown an association between e-cigarette use and subsequent smoking among nonsmokers, which suggests the possibility that e-cigarette use may also act as a gateway to smoking (sometimes referred to as the gateway hypothesis), particularly among adolescents. A recent meta-analysis found that for young people aged 30 years or younger, there is a strong and consistent positive association between e-cigarette use among never smokers and later smoking, but that there is currently insufficient evidence to conclude that this association is causal [5]. Understanding more about the nature of the association between smoking and e-cigarette use, particularly in young adulthood, is vital to inform tobacco control policies that aim to prevent youth smoking initiation by restricting access to e-cigarettes. Specifically, it is important to understand whether the association found among young adults is causal, or due to other factors that influence both smoking and e-cigarette use independently.

For example, there is some evidence for a shared genetic liability to both smoking and e-cigarette use [6]. This could indicate a causal relationship in that genetic variants influence smoking which then increases the probability of e-cigarette use (i.e., vertical pleiotropy), or it could be due to genetic variants that influence a phenotype which consequently influences both behaviours (i.e., horizontal pleiotropy) [7]. One biologically plausible explanation for a genetic link between smoking and e-cigarette use is that they are both influenced by the same genetic variants that influence an individual's response to nicotine or their nicotine metabolism. However, evidence suggests that some of the genetic influence on smoking initiation is mediated by personality traits, such as risk-taking and impulsivity, that influence (among other things) smoking uptake [8]. Allegrini and colleagues [6] suggest that a genetic link between smoking and e-cigarette use may reflect these personality traits (i.e., a genetic liability to take risks may influence an individual's likelihood of initiating smoking and e-cigarette use).

Using genetic variants, we can explore whether smoking is associated with e-cigarette use, and which factors or mechanisms may influence the association. Ideally, we would explore the genetic overlap between smoking and e-cigarette use by comparing the genetic variants identified in genome-wide association studies (GWAS) of each behaviour, but at present, there are no large, well-powered GWAS of e-cigarette use. However, a GWAS of various smoking behaviours has recently been published [9], which identified 378 single nucleotide polymorphisms (SNPs) associated with smoking initiation. Using these SNPs, smoking initiation polygenic risk scores (PRS) can be created and associations between these PRS and a range of outcomes examined.

Causality cannot be inferred from such analyses, but negative control outcomes can be used to inform the overall evaluation of whether an association is causal via a hypothesised route. Negative controls are outcomes which are not plausibly caused by the exposure—for example, smoking is associated with risk of dying by suicide (which is biologically plausible), but equally strongly associated with risk of dying by homicide (which is not), casting doubt on the causal nature of the former association [10]. Triangulating evidence from outcomes where a simple biological pathway from smoking to the outcome is implausible (e.g., gambling), or impossible (e.g., externalising behaviour or socioeconomic position [SEP] in childhood, before smoking has occurred) can aid consideration of potential pathways by which smoking and e-cigarette use may share a genetic predisposition. These potential pathways (displayed in Fig 1) include a

## a) Vertical pleiotropy

Genetic predisposition ⟶ Smoking initiation ⟶ E-cigarette use

## b) Horizontal pleiotropy

Genetic predisposition ⟶ Smoking initiation
Genetic predisposition ⟶ E-cigarette use

## c) Common risk factor

Genetic predisposition ⟶ Risk-taking ⟶ Smoking initiation
Risk-taking ⟶ E-cigarette use

**Fig 1. Potential models of shared liability for the relationship between genetic predisposition to smoking initiation and e-cigarette use.**

biological pathway from smoking to e-cigarette use (i.e., vertical pleiotropy), a shared genetic predisposition which influences smoking and e-cigarette use independently (i.e., horizontal pleiotropy), or a genetic liability to a broader, risk-taking phenotype (i.e., a shared risk factor) which causes both smoking and e-cigarette use. Alternatively, triangulation could help us consider whether an association is due to a shared genetic predisposition between parents and offspring. Where parents share their offspring's smoking initiation predisposition, they are likely to expose their offspring to cigarette smoke in utero or in childhood. Consequently, an apparent effect of a child's own genetic variants may be a result of their prenatal or postnatal environment due to a dynastic effect of their parents' genetic variants. If associations are only found between smoking initiation PRS and e-cigarette use, but not negative control outcomes, this would strengthen the vertical pleiotropy interpretation; however, if an association is also found with negative control outcomes, this would indicate that horizontal pleiotropy is occurring or that shared parent–offspring genetic predisposition may be confounding the association.

Additionally, using varying p-value thresholds to create PRS could help to identify the presence of horizontal pleiotropy. Calculating PRS at less strict p-value thresholds than the standard genome-wide significant threshold increases the percentage variance in the phenotype explained by the score, and thus increases power to detect an association. However, using less stringent thresholds will also tend to increase the likelihood of including genetic variants which are related to other factors, making the PRS less specific to the exposure of interest (and may eventually result in PRS which explain less variance in the exposure). The more SNPs

included in a PRS, the less likely it is that the effect of each variant on the trait of interest is proportional to the effect of the trait of interest on the exposure, and the more likely it is proportional to the effects on other (horizontally pleiotropic) traits [11], increasing the likelihood that any associations found between the PRS and an outcome could be due to horizontal pleiotropy. Triangulating evidence from a variety of thresholds and a variety of outcomes may provide a clearer picture of the true association. Associations observed when more stringent PRS thresholds are used could be due to a causal effect of smoking, and consistent magnitudes of association at less stringent thresholds could indicate that any associations observed are driven by the effect of the more specific PRS. However, increasing magnitudes of association observed at less stringent thresholds (particularly among negative control outcomes) may indicate horizontal pleiotropy is driving part of the associations observed.

We aimed to investigate whether smoking initiation PRS are associated with ever use of e-cigarettes in young adulthood. Given the possibility of a shared liability mechanism (e.g., an underlying risk-taking phenotype), we also aimed to explore any associations with outcomes that are not plausibly biologically related (e.g., gambling) or that precede smoking (e.g., hyperactivity in childhood), to determine whether the association between smoking and e-cigarette use could reflect a broader risk-taking phenotype captured by the smoking initiation PRS (i.e., a common risk factor). Finally, we aimed to explore whether the smoking initiation PRS may be capturing broader social influences on smoking (e.g., socioeconomic position at birth) which cannot plausibly have been a causal effect of the young adult's own smoking.

## Methods

This study is reported as per the Strengthening the Reporting of Observational Studies in Epidemiology (STROBE) guideline (S1 STROBE Checklist).

### Data sources

Two data sources were utilised for this study: the GWAS & Sequencing Consortium of Alcohol and Nicotine use (the discovery sample; GSCAN) and the Avon Longitudinal Study of Parents and Children (the target sample; ALSPAC) [9,12,13].

**GSCAN.** GSCAN report summary level statistics from a GWAS of smoking initiation [9]. This GWAS was based on 1,232,091 participants from 29 cohorts. In order to eliminate data overlap with the target sample, summary statistics were obtained (through correspondence with GSCAN) with ALSPAC participants removed ($N = 11,345$). 23andMe participants ($N = 599,289$) were also excluded from this summary data due to data sharing restrictions. The remaining summary data consisted of data from 621,457 participants. Smoking initiation was defined as ever being a regular smoker. The exact definition varied across the cohorts included in the consortium, with 3 different definitions: (1) Have you smoked over 100 cigarettes over the course of your life? (2) Have you ever smoked every day for at least a month? (3) Have you ever smoked regularly? The majority of the SNPs identified were intergenic with no known function, but glutamate and dopaminergic gene pathways were enriched for smoking initiation. Also, the rs6265 variant (a nonsynonymous SNP in the *BDNF* gene) which has previously been found to be associated with smoking initiation [14] was also associated with smoking initiation in GSCAN. A comprehensive description of the genetic variants, and the genes they are within (e.g., *PPP1R1B*, *GRIN2A*, *HOMER2*), have been described previously [9].

**ALSPAC.** The target sample consisted of participants from ALSPAC [12,13], a prospective cohort study with extensive data from birth to young adulthood (including genetic data). This study recruited pregnant women residing in Avon, UK with expected delivery dates between 1 April 1991 and 31 December 1992. The phases of enrolment are described in detail in the cohort

profile paper and its update [15]. A total of 15,454 mothers were recruited, resulting in 15,589 foetuses. Of these, 14,901 children were alive at 1 year of age. The study website contains details of all the data that are available via a fully searchable data dictionary and variable search tool (http://www.bristol.ac.uk/alspac/researchers/our-data/). After samples that did not pass quality control were removed, genetic data were available for 9,085 young adults. PRS were created for 7,859 unrelated individuals of European ancestry. Of these individuals, 2,905 also had data for our main outcome (e-cigarette use) at 24 years (the most recent time point at which detailed e-cigarette use data were collected prior to analysis). ALSPAC study data from 22 years onwards were collected and managed using REDCap electronic data capture tools hosted at the University of Bristol [16]. Sample sizes varied by outcome due to restrictions (e.g., restricting to never smokers) and differing time points of measurement (i.e., missing data).

## Ethics

Ethics approval for the study was obtained from the ALSPAC Ethics and Law Committee and the Local Research Ethics Committees. Consent for biological samples has been collected in accordance with the Human Tissue Act (2004). Written informed consent for the use of data collected via questionnaires and clinics was obtained from participants following the recommendations of the ALSPAC Ethics and Law Committee at the time of study initiation (i.e., 1991).

## Polygenic risk scores

Summary data from GSCAN (excluding ALSPAC and 23andMe, $N = 621,457$) were used to select SNPs associated with smoking initiation. Betas were converted to log odds ratios (ORs). Each participant was given a score which indicated the average number of risk alleles (0, 1, or 2 effect alleles) they possessed for the selected SNPs. Scores were weighted (i.e., multiplied) by the regression coefficients from the summary statistics (with ALSPAC and 23andMe removed), then standardised by transforming to z-scores. Five $p$-value thresholds ($5 \times 10^{-8}$, 0.0005, 0.005, 0.05, 0.5) were used to determine 5 groups of SNPs to be included in 5 different PRS for each participant. PLINK was used to determine PRS at the $p < 5 \times 10^{-8}$ threshold using the SNPs which met the genome-wide significance threshold in the GSCAN GWAS of smoking initiation [9]. PRSice software was used to calculate the PRS at all other thresholds [17]. The data acquired from GSCAN was pruned for SNPs with a Minor Allele Frequency (MAF) > 0.001 where at least 10% of the maximum sample size had SNP data available in at least 3 of the consortium studies. SNPs were clumped to ensure low linkage disequilibrium ($r^2 < 0.1$).

## Outcomes

Detailed information regarding the phenotype data including the questions and answer options provided in the questionnaires are available in S1 Table.

**E-cigarette use.** At 24 years (between 2016 and 2017), outcome data were collected via questionnaire on whether participants had ever used e-cigarettes. Ever use was defined as ever having used/vaped an e-cigarette or other vaping device.

**Smoking.** Self-reported smoking initiation and ever smoking were included as positive control outcomes (i.e., outcomes for which an association with the exposure is expected). Smoking initiation by 24 years was defined as having smoked 100 or more cigarettes in their lifetime. Ever smoking by 24 years was defined as having ever smoked a whole cigarette (including roll-ups).

**Negative controls.** Four negative control outcomes at age 23 and 24 were included in the analysis: high number of sexual partners, having been in trouble with the law, ever gambling, and enjoying taking risks. These were selected on the basis of being related to broad risk-taking

behaviour, but where a causal pathway from smoking was not considered biologically plausible. Three negative control outcomes at age 7 were included: hyperactivity, conduct disorder (CD), and oppositional defiant disorder (ODD). These externalising disorders are indicators of impulsivity and were selected on the basis that few (if any) children at this age have smoked, ruling out a causal pathway from their own smoking to these outcomes. Parental SEP, which was measured at birth, was also included in the analysis. This outcome was based on highest occupation of both parents at birth (preceding smoking) and was selected on the basis that it could not possibly be caused by a young person's own smoking. Further information regarding the negative controls can be found in S1 Text.

## Statistical analyses

After creating the PRS using PLINK and PRSice software, all analyses were carried out in STATA 15.1 [18]. Using the logistic command, we conducted a series of logistic regressions adjusted for age (in months at the time of the outcome measure), sex, and the first 10 principal components of population stratification (i.e., common subpopulation differences in allele frequencies). We assessed the association between smoking initiation PRS and (i) ever e-cigarette use by age 24 among the full sample and those who had never smoked, (ii) regular e-cigarette use at age 24, (iii) smoking initiation, and (iv) negative control outcomes (risk-taking behaviours, externalising disorders, and SEP). All analyses were repeated for each of the 5 $p$-value thresholds for determining SNP inclusion in the PRS. We also assessed the association between the main outcome of interest (e-cigarette use) and each negative control outcome. These analyses were planned prior to the analysis being conducted and were not data driven; however, the plan was not made publicly available prior to the analysis.

## Results

A total of 378 SNPs were identified as genome-wide significant in the GSCAN GWAS of smoking initiation [9], 356 of which were available in ALSPAC. Nine SNPs were removed at the clumping stage, leaving 347 SNPs included in the most stringent PRS ($p$-value threshold $p < 5 \times 10^{-8}$). The number of SNPs included in each PRS at the less stringent thresholds is shown in S2 Table. Of note, PRS calculated at these less stringent thresholds were based on the significance level reported in the restricted sample (excluding ALSPAC and 23andMe) summary data.

Table 1 shows the characteristics of the sample; 878 (30%) young adults were self-reported ever e-cigarette users by 24 years, and 1,695 (64%) were self-reported ever smokers. Of those who had ever used an e-cigarette, 95% ($n = 830$) had ever smoked at least one whole cigarette, and 71% ($n = 616$) had smoked 100 or more cigarettes. Less than 1% of the sample had used an e-cigarette prior to smoking. Self-reported smoking and e-cigarette use were associated with lower parental SEP and having externalising disorders in childhood (S3 Table). Self-reported smoking and e-cigarette use were also associated with increased odds of engaging in risk-taking behaviours (S3 Table).

## Smoking initiation PRS and self-reported smoking

We observed positive associations between smoking initiation PRS and ever smoking (having smoked at least 1 cigarette in a lifetime) by the age of 24 years ($p < 5 \times 10^{-8}$ threshold OR ($OR_{10-8}$) = 1.25, 95% CI 1.16 to 1.35, $p < 0.001$) and smoking initiation (having smoked at least 100 cigarettes in a lifetime) by the age of 24 years ($OR_{10-8}$ = 1.29, 95% CI 1.19 to 1.39, $p < 0.001$). We found strong associations between smoking initiation PRS and self-reported smoking measures at all $p$-value thresholds (Table 2).

**Table 1. Characteristics of young adults in ALSPAC.**

| Characteristic | N (%) |
|---|---|
| Ever used an e-cigarette by 24 (used once or more) | 878 (30%) |
| Regularly used an e-cigarette at 24 (used at least once a month) | 150 (5%) |
| Ever smoked by 24 (1 cigarette or more) | 1,695 (64%) |
| Initiated smoking by 24 (100 cigarettes or more) | 972 (33%) |
| Ever used an e-cigarette but not initiated smoking by 24 | 262 (13%) |
| High number of sexual partners at 23* | 647 (25%) |
| Been in trouble with the law since 23rd birthday | 69 (2%) |
| Enjoys taking risks at 24 | 1,618 (55%) |
| Ever gambled at 24 | 2,156 (74%) |
| Hyperactivity at 7 | 2,219 (42%) |
| Conduct disorder at 7 | 1,199 (22%) |
| Oppositional defiant disorder at 7 | 1,868 (35%) |
| Parental SEP (manual) | 1,068 (27%) |
| | Mean (SD) |
| Age in months at 24-year questionnaire | 298 (6) |

Sample sizes varied by characteristic due to differing time points of measurement (i.e., missing data).

*Eleven or more sexual partners, determined using the upper quartile for number of lifetime sexual partners in the ALSPAC sample (11 sexual partners).

ALSPAC, Avon Longitudinal Study of Parents and Children; SEP, socioeconomic position.

## Smoking initiation PRS and self-reported e-cigarette use

We observed positive associations between smoking initiation PRS and self-reported ever use of e-cigarettes by the age of 24 years ($OR_{10-8}$ = 1.24, 95% CI 1.14 to 1.34, $p < 0.001$) and self-reported regular (at least once a month) e-cigarette use at 24 years ($OR_{10-8}$ = 1.18, 95% CI 1.00 to 1.40, $p = 0.049$). We observed these associations at all $p$-value thresholds (Table 2). Among those who had never initiated smoking (i.e., smoked <100 cigarettes in their lifetime), we found no clear evidence for an association between smoking initiation PRS and ever e-cigarette use at the most stringent $p$-value thresholds. However, we found evidence of a positive association with PRS calculated using less stringent thresholds ($p < 0.5$ threshold OR = 1.18, 95% CI 1.04 to 1.35, $p = 0.012$; Table 2). We found similar patterns of association among those who had never smoked any cigarettes (S4 Table).

## Smoking initiation PRS and negative controls

We observed a positive association between smoking initiation PRS and high number of sexual partners by 23 years ($OR_{10-8}$ = 1.15, 95% CI 1.05 to 1.26, $p = 0.003$) and having ever gambled by 24 years ($OR_{10-8}$ = 1.12, 95% CI 1.03 to 1.22, $p = 0.008$) at all $p$-value thresholds (Table 3). We found some evidence of a positive association between smoking initiation PRS and enjoying taking risks at 24 years ($OR_{0.005}$ = 1.11, 95% CI 1.03 to 1.19, $p = 0.005$), but this was less clear at the more stringent thresholds (Table 3). There was no clear evidence of an association between smoking initiation PRS and having been in trouble with the law since their 23rd birthday (Table 3).

We found evidence of a positive association between smoking initiation PRS and hyperactivity at 7 years ($OR_{0.0005}$ = 1.10, 95% CI 1.04 to 1.16, $p = 0.001$) but not at the most stringent threshold (Table 4). There was also a positive association with CD at 7 years ($OR_{10-8}$ = 1.10, 95% CI 1.03 to 1.17, $p = 0.004$) at all thresholds (Table 4). There was some evidence of a

**Table 2. Associations between polygenic risk scores for smoking initiation with ever e-cigarette use, ever smoking, and smoking initiation.**

| Outcome _p_-value threshold | _n_ | OR | 95% CI | _p_ |
|---|---|---|---|---|
| Ever e-cigarette use by age 24 | 2,894 | | | |
| $5 \times 10^{-8}$ | | 1.24 | 1.14, 1.34 | <0.001 |
| 0.0005 | | 1.27 | 1.17, 1.38 | <0.001 |
| 0.005 | | 1.36 | 1.26, 1.48 | <0.001 |
| 0.05 | | 1.39 | 1.28, 1.51 | <0.001 |
| 0.5 | | 1.39 | 1.28, 1.51 | <0.001 |
| Regular e-cigarette use at age 24 (at least once a month) | 2,894 | | | |
| $5 \times 10^{-8}$ | | 1.18 | 1.00, 1.40 | 0.049 |
| 0.0005 | | 1.22 | 1.03, 1.44 | 0.019 |
| 0.005 | | 1.22 | 1.04, 1.44 | 0.017 |
| 0.05 | | 1.18 | 1.00, 1.39 | 0.051 |
| 0.5 | | 1.22 | 1.04, 1.44 | 0.018 |
| Ever smoking by age 24 (1 cigarette or more) | 2,931 | | | |
| $5 \times 10^{-8}$ | | 1.25 | 1.16, 1.35 | <0.001 |
| 0.0005 | | 1.27 | 1.17, 1.38 | <0.001 |
| 0.005 | | 1.32 | 1.22, 1.43 | <0.001 |
| 0.05 | | 1.33 | 1.23, 1.44 | <0.001 |
| 0.5 | | 1.34 | 1.24, 1.44 | <0.001 |
| Smoking initiation (100 cigarettes or more) by age 24 | 2,925 | | | |
| $5 \times 10^{-8}$ | | 1.29 | 1.19, 1.39 | <0.001 |
| 0.0005 | | 1.38 | 1.27, 1.49 | <0.001 |
| 0.005 | | 1.46 | 1.34, 1.58 | <0.001 |
| 0.05 | | 1.49 | 1.37, 1.61 | <0.001 |
| 0.5 | | 1.49 | 1.37, 1.39 | <0.001 |
| Ever e-cigarette use by age 24 among never smokers (<100 cigarettes) | 1,937 | | | |
| $5 \times 10^{-8}$ | | 1.10 | 0.97, 1.26 | 0.150 |
| 0.0005 | | 1.05 | 0.92, 1.20 | 0.464 |
| 0.005 | | 1.12 | 0.98, 1.28 | 0.087 |
| 0.05 | | 1.15 | 1.00, 1.31 | 0.046 |
| 0.5 | | 1.18 | 1.04, 1.35 | 0.012 |

Ever smoking and smoking initiation models were included as positive controls. Analyses were adjusted for age, sex, and principal components 1–10.

OR, odds ratio.

positive association between PRS and ODD specifically at the 0.0005 threshold (OR$_{0.0005}$ = 1.08, 95% CI 1.02 to 1.14, _p_ = 0.013). We also found a positive association with lower parental SEP (OR$_{10-8}$ = 1.08, 95% CI 1.01 to 1.16 _p_ = 0.017) at all thresholds (Table 5).

## Discussion

In this study, we explored the association between smoking initiation PRS and e-cigarette use, using logistic regression. We further explored the findings by observing the association between smoking initiation PRS and positive controls (smoking) and negative controls (e.g., risk-taking), as well as restricting the analysis to never smokers. Smoking initiation PRS were strongly associated with ever e-cigarette use by 24 years whereby higher genetic liability to smoking initiation was associated with a 24% increase in the likelihood of ever using an e-

**Table 3. Associations between polygenic risk scores for smoking initiation with negative controls of risky behaviour.**

| Outcome<br>p-value threshold | n | OR | 95% CI | p |
|---|---|---|---|---|
| 11 or more sexual partners by age 23* | 2,505 | | | |
| $5 \times 10^{-8}$ | | 1.15 | 1.05, 1.26 | 0.003 |
| 0.0005 | | 1.12 | 1.02, 1.23 | 0.019 |
| 0.005 | | 1.18 | 1.08, 1.29 | <0.001 |
| 0.05 | | 1.25 | 1.14, 1.37 | <0.001 |
| 0.5 | | 1.30 | 1.19, 1.43 | <0.001 |
| Been in trouble with the law since 23rd birthday | 2,928 | | | |
| $5 \times 10^{-8}$ | | 1.00 | 0.79, 1.28 | 0.988 |
| 0.0005 | | 1.12 | 0.88, 1.43 | 0.352 |
| 0.005 | | 1.11 | 0.87, 1.41 | 0.407 |
| 0.05 | | 1.04 | 0.82, 1.33 | 0.745 |
| 0.5 | | 0.90 | 0.71, 1.15 | 0.394 |
| Enjoys taking risks at age 24 | 2,932 | | | |
| $5 \times 10^{-8}$ | | 1.06 | 0.98, 1.14 | 0.154 |
| 0.0005 | | 1.05 | 0.98, 1.14 | 0.163 |
| 0.005 | | 1.11 | 1.03, 1.19 | 0.005 |
| 0.05 | | 1.09 | 1.01, 1.17 | 0.029 |
| 0.5 | | 1.08 | 1.01, 1.16 | 0.033 |
| Ever gambled by age 24 | 2,899 | | | |
| $5 \times 10^{-8}$ | | 1.12 | 1.03, 1.22 | 0.008 |
| 0.0005 | | 1.16 | 1.07, 1.26 | 0.001 |
| 0.005 | | 1.16 | 1.06, 1.26 | 0.001 |
| 0.05 | | 1.20 | 1.10, 1.30 | <0.001 |
| 0.5 | | 1.15 | 1.06, 1.25 | 0.001 |

Number of sexual partners, trouble with the law, enjoying risk-taking, and gambling models were included as negative controls. Analyses were adjusted for age, sex, and principal components 1–10.

*Low (<11) vs. high (11 or more) number of sexual partners, determined using the upper quartile for number of lifetime sexual partners in the ALSPAC sample (11 sexual partners).

ALSPAC, Avon Longitudinal Study of Parents and Children; OR, odds ratio.

cigarette (per standard deviation (SD) increase in PRS). As expected, we observed an association of smoking initiation PRS and both ever smoking and smoking initiation. It was notable that the associations of the smoking initiation PRS and both smoking and e-cigarette use were of similar magnitude. Given the small amount of variation in smoking initiation explained by the SNPs (2.3%), and the fact that any causal effect will only explain a proportion of the outcome, these small effect sizes are to be expected. Interestingly, we also observed positive associations between smoking initiation PRS and risk-taking, impulsivity, and parental SEP at birth.

In contrast to the results of Allegrini and colleagues [6], we found an association between ever e-cigarette use and smoking initiation predisposition where they only found an association with smoking heaviness predisposition. This is likely due to the use of different SNPs to create the score; our score was based on the findings of a recent, large GWAS (GSCAN [9]; N = 1,232,091), whereas Allegrini and colleagues [6] based their score on an earlier GWAS with a much smaller sample (the Tobacco and Genetics Consortium [14]; N = 69,207). Thus, there was greater statistical power to detect genome-wide significant associations in the GWAS we based our score on.

**Table 4. Associations between polygenic risk scores for smoking initiation with negative controls of externalising disorders in childhood.**

| Outcome<br>*p*-value threshold | *n* | OR | 95% CI | *p* |
|---|---|---|---|---|
| Hyperactivity at age 7 | 5,227 | | | |
| $5 \times 10^{-8}$ | | 1.02 | 0.96, 1.08 | 0.511 |
| 0.0005 | | 1.10 | 1.04, 1.16 | 0.001 |
| 0.005 | | 1.14 | 1.08, 1.20 | <0.001 |
| 0.05 | | 1.14 | 1.08, 1.21 | <0.001 |
| 0.5 | | 1.15 | 1.08, 1.21 | <0.001 |
| Conduct disorder at age 7 | 5,334 | | | |
| $5 \times 10^{-8}$ | | 1.10 | 1.03, 1.17 | 0.004 |
| 0.0005 | | 1.11 | 1.04, 1.19 | 0.001 |
| 0.005 | | 1.11 | 1.04, 1.18 | 0.002 |
| 0.05 | | 1.08 | 1.01, 1.15 | 0.021 |
| 0.5 | | 1.08 | 1.01, 1.15 | 0.017 |
| Oppositional defiant disorder at age 7 | 5,325 | | | |
| $5 \times 10^{-8}$ | | 1.02 | 0.96, 1.08 | 0.496 |
| 0.0005 | | 1.08 | 1.02, 1.14 | 0.013 |
| 0.005 | | 1.04 | 0.98, 1.10 | 0.200 |
| 0.05 | | 1.04 | 0.98, 1.10 | 0.173 |
| 0.5 | | 1.02 | 0.96, 1.08 | 0.529 |

Hyperactivity and conduct disorder were assessed using the strengths and difficulties questionnaire (SDQ), and oppositional defiant disorder was assessed using the development and wellbeing assessment (DAWBA). All variables were recoded into binary outcomes (no disorder/symptoms versus borderline/disorder/symptoms).
DAWBA, development and wellbeing assessment; OR, odds ratio; SDQ, strengths and difficulties questionnaire.

The association between smoking initiation PRS and e-cigarette use could be explained by smoking causally influencing e-cigarette use. This hypothesis is supported by observational evidence; use of e-cigarettes for smoking cessation is common among both young adults in the UK [4] and adults in Great Britain [19]. However, the associations observed among the restricted analysis and between the negative control outcomes suggest there may be other factors at play—there may be shared genetic risk factors that influence both behaviours. Among

**Table 5. Associations between polygenic risk scores for smoking initiation with negative controls of socioeconomic indicators.**

| Outcome<br>*p*-value threshold | *n* | OR | 95% CI | *p* |
|---|---|---|---|---|
| Parental SEP (manual) | 6,702 | | | |
| $5 \times 10^{-8}$ | | 1.08 | 1.01, 1.16 | 0.017 |
| 0.0005 | | 1.13 | 1.06, 1.21 | <0.001 |
| 0.005 | | 1.16 | 1.09, 1.24 | <0.001 |
| 0.05 | | 1.11 | 1.03, 1.18 | 0.003 |
| 0.5 | | 1.13 | 1.05, 1.20 | <0.001 |

Parental SEP was based on the higher of the mother or partner's occupational social class using the Office of Population Censuses and Surveys (OPCS) occupation codes.
OPCS, Office of Population Censuses and Surveys; OR, odds ratio; SEP, socioeconomic position.

never smokers, we found weak evidence of an association between smoking initiation PRS and e-cigarette use, which suggests that the e-cigarette use is not simply caused by smoking (which has not occurred in these cases) but that there is a shared genetic aetiology influencing both behaviours. Hence, what appears to be a gateway between e-cigarette use and smoking in previous studies [5] could actually be a shared genetic liability, and the order of use is coincidental or due to other factors such as perceived risk or misreporting of smoking status [20].

Alternatively, the smoking initiation PRS may be capturing much more than just smoking or nicotine use. Using less stringent *p*-value thresholds to create PRS increases the percentage variance in the phenotype explained by the score, and therefore the power to detect an association up to a point; using less stringent thresholds also increases the likelihood of capturing SNPs which are related to other factors, which adds noise and eventually results in less specific PRS that explain less variance in the exposure and more variance in other (horizontally pleiotropic) effects. Increasing magnitudes and strengthened evidence of association with PRS and negative controls at less stringent *p*-value thresholds suggests that the smoking initiation PRS is capturing, at least in part, a broad phenotype which is not entirely specific to smoking/nicotine. Although weaker associations were observed between risk-taking factors and PRS for smoking initiation compared to e-cigarette use and smoking, the associations are still relatively strong and consistent. Recent observational evidence also indicated a strong association between e-cigarette use and smoking prior to adjusting for risk-taking behaviours and other shared risk factors but showed no clear evidence of an association after adjusting for risk-taking behaviours and other shared risk factors [21]. We also found an association between the smoking initiation PRS and externalising disorders in childhood (7 years) which precedes the age at which cigarettes are first smoked in the vast majority of cases in this cohort (>99%) and therefore cannot be a causal effect of own smoking. However, this association could potentially be due to causal in utero effects of maternal smoking in pregnancy or maternal smoking in childhood, since maternal and offspring genotype will be correlated. Nevertheless, combined with evidence that liability to attention-deficit/hyperactivity disorder (ADHD) increases the likelihood of smoking initiation and vice versa [22], our results suggest the possibility that the smoking initiation PRS is capturing a broad impulsivity phenotype. The association observed between PRS for smoking initiation and parental SEP also suggests the PRS could be capturing sociodemographic factors as well as smoking. Alternatively, there may be a dynastic effect whereby parental predisposition to smoking (which is correlated with their child's genetic predisposition) influences parental SEP at birth. The apparent association of the child's genotype could actually be an outcome of parental genetic predisposition,

Despite the strengths of this study (which include the use of a well-powered GWAS to create our score, lack of overlap between samples, and use of negative controls to explore potential mechanistic pathways), there are a number of limitations of this study. First, the relatively low sample size—particularly when investigating associations with regular e-cigarette use and restricting to never smokers. Second, restricting analysis to never smokers could introduce collider bias [23]. We found that smoking initiation PRS were strongly associated with smoking initiation; if e-cigarette use causes young adults to smoke, then smoking status is a collider and conditioning on this variable (i.e., restricting analysis to never smokers) may inflate any association between smoking initiation PRS and e-cigarette use. Third, this cohort is not appropriate to directly study the gateway hypothesis as the young adults in ALSPAC were approximately 17 years old when e-cigarettes became widely available and therefore were exposed to cigarettes earlier in their adolescence than e-cigarettes and had more opportunity to smoke than use e-cigarettes than later birth cohorts. Future research should explore this association in a larger sample of individuals with exposure to both cigarettes and e-cigarettes during adolescence. Fourth, the attrition rate in ALSPAC is considerable—only 2,905 of the 7,859

nonrelated participants of European ancestry with genetic data responded to the questions about e-cigarette use in the 24 year questionnaire—and missingness in this cohort has previously been associated with smoking initiation PRS [24]. Replicating the participation scores used by Taylor and colleagues [24], we found that higher smoking initiation PRS were associated with participating in fewer ALSPAC questionnaires and clinics (change in participation per SD increase in smoking initiation PRS [$p < 5 \times 10^{-8}$ threshold] = −1.15, 95% CI −1.53 to −0.76, $p < 0.001$). Furthermore, we found that those with higher smoking initiation PRS were less likely to have been included in the analysis of smoking initiation PRS and e-cigarette use due to attrition ($OR_{10\text{-}8}$ per SD of smoking initiation PRS = 0.87, 95% CI 0.83 to 0.91, $p < 0.001$) so our estimates may be biased by selection and the association could be stronger than observed here. However, interpretation of any study including smoking initiation PRS will be difficult as the association between smoking initiation PRS and attrition could induce bias such as collider bias [25]. Fifth, the variability in the nature of the key assessments and the use of self-reports may have resulted in measurement error of the phenotype and outcomes.

The associations observed here may have implications for the use of smoking initiation PRS in mendelian randomisation (MR) analysis. This method is often implemented to provide unconfounded causal estimates, as long as the assumptions of MR hold [26]. One assumption is that the genetic instrument (e.g., smoking initiation PRS) is not associated with any confounders (e.g., risk-taking, childhood externalising disorders, SEP). The association we observed between smoking initiation PRS and negative control outcomes, even when restricted to only genome-wide significant SNPs, indicates that smoking initiation PRS may not be a valid instrument to use in MR to investigate the causal effects of smoking initiation. This emphasises the importance of using pleiotropy robust methods (e.g., MR Egger). The InSIDE (Instrument Strength Independent of Direct Effect) assumption requires that SNP-exposure effects (e.g., the effect of smoking initiation SNPs on smoking initiation) should not be correlated with horizontal pleiotropic effects (e.g., the effect of smoking initiation SNPs on broad risk-taking behaviour). The association observed between the smoking initiation PRS and multiple risk-taking behaviours and externalising disorders in childhood suggests that the smoking initiation SNPs may be capturing a broader phenotype, such as risk-taking, which is not specific to smoking or nicotine, and thus this assumption may be violated. One approach which could be used to address this is Steiger filtering which can be used to exclude SNPs which explain the variance in the outcome over and above the variance in the exposure [11,27]. The same approach can be applied in MR studies using smoking initiation PRS to remove SNPs which explain more variance in the negative control outcomes used in this study (or other phenotypes/proxies for risk-taking behaviour) than variance in smoking initiation. However, if the InSIDE assumption is perfectly violated (i.e., if the SNP effect on broad risk-taking causes smoking initiation), the smoking initiation PRS will be an invalid instrument using any MR method. At the very least, triangulating evidence across multiple MR methods (e.g., median weighted and mode based) would be advised in MR studies using smoking initiation PRS but, ideally, other causal inference methods should also be used. Further research could also explore the potential mediating effects of the positive and negative controls included in this analysis; if a PRS for e-cigarette initiation is identified in a GWAS, pleiotropy robust multivariable MR methods [28] could be employed to explore mediating effects using smoking initiation, e-cigarette initiation, and risk-taking PRS (providing the PRS are sufficiently independent from one another).

The results also provide support for a shared genetic liability between e-cigarette use and smoking, which may have implications for policy; strict policies (e.g., bans), which aim to prevent e-cigarette use in order to reduce the risk of smoking initiation among youth and young adults, may not be effective. In fact, they may have the opposite effect; if young people are

predisposed to both e-cigarette use and smoking but only cigarettes are available, this could increase their likelihood of smoking because it is the only option available to them. Furthermore, such policies may prevent and discourage adult smokers from accessing an effective smoking cessation tool and hamper smoking cessation attempts and could therefore have a negative impact on smoking rates. Ideally, policy should prevent use by nonsmokers but promote use by smokers for smoking cessation.

In conclusion, we find evidence to suggest there is a shared genetic aetiology between smoking and e-cigarette use but also with risky behaviour, SEP, and externalising disorders in childhood. This suggests the PRS for smoking initiation is not specific to smoking or nicotine use but is capturing something much broader. Future research is needed to explore this in a population which has been exposed to both e-cigarettes and cigarettes in adolescence.

## Supporting information

**S1 STROBE Checklist. STROBE Checklist.**
(DOCX)

**S1 Text. Supplementary material.**
(DOCX)

**S1 Table. Questionnaire items and possible responses.**
(DOCX)

**S2 Table. *p*-Value thresholds and number of SNPs included in polygenic risk scores.**
(DOCX)

**S3 Table. Association between self-reported e-cigarette use and smoking and risk-taking behaviours, socioeconomic indicators, and externalising disorders in childhood.**
(DOCX)

**S4 Table. Association between polygenic risk scores for smoking initiation with ever e-cigarette use among never smokers (smoked <1 cigarette in their lifetime).**
(DOCX)

## Acknowledgments

We are extremely grateful to all the families who took part in this study, the midwives for their help in recruiting them, and the whole ALSPAC team, which includes interviewers, computer and laboratory technicians, clerical workers, research scientists, volunteers, managers, receptionists, and nurses.

The views expressed in this publication are those of the authors and not necessarily those of the NHS, the National Institute for Health Research, or the Department of Health and Social Care.

## Author Contributions

**Conceptualization:** Jasmine N. Khouja, George Davey Smith, Marcus R. Munafò.

**Data curation:** Jasmine N. Khouja.

**Formal analysis:** Jasmine N. Khouja, Robyn E. Wootton.

**Funding acquisition:** Amy E. Taylor, Marcus R. Munafò.

**Investigation:** Jasmine N. Khouja.

**Methodology:** Jasmine N. Khouja, Robyn E. Wootton, Amy E. Taylor, George Davey Smith, Marcus R. Munafò.

**Project administration:** Jasmine N. Khouja.

**Resources:** Robyn E. Wootton, Amy E. Taylor, Marcus R. Munafò.

**Supervision:** Robyn E. Wootton, Amy E. Taylor, Marcus R. Munafò.

**Writing – original draft:** Jasmine N. Khouja.

**Writing – review & editing:** Jasmine N. Khouja, Robyn E. Wootton, Amy E. Taylor, George Davey Smith, Marcus R. Munafò.

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
