## [Editor Report · Decision Letter 0]

22 Jun 2020

Dear Dr Khouja, 

Thank you for submitting your manuscript entitled "Association of genetic liability to smoking initiation with e-cigarette use in young adults." for consideration by PLOS Medicine.

Your manuscript has now been evaluated by the PLOS Medicine editorial staff as well as by an academic editor with relevant expertise and I am writing to let you know that we would like to send your submission out for external peer review.

Kind regards,

Thomas J McBride, PhD,

PLOS Medicine

---

## [Decision Letter · Decision Letter 1]

16 Sep 2020

Dear Dr. Khouja,

Thank you very much for submitting your manuscript "Association of genetic liability to smoking initiation with e-cigarette use in young adults." (PMEDICINE-D-20-02674R1) for consideration at PLOS Medicine. 

Your paper was evaluated by a senior editor and discussed among all the editors here. It was also discussed with an academic editor with relevant expertise, and sent to four independent reviewers, including a statistical reviewer. The reviews are appended at the bottom of this email and the accompanying attachment from reviewer 4 can be seen via the link below:

[LINK]

In light of these reviews, I am afraid that we will not be able to accept the manuscript for publication in the journal in its current form, but we would like to consider a revised version that addresses the reviewers' and editors' comments. Obviously we cannot make any decision about publication until we have seen the revised manuscript and your response, and we plan to seek re-review by one or more of the reviewers. 

We expect to receive your revised manuscript by Oct 07 2020 11:59PM. Please email us (plosmedicine@plos.org) if you have any questions or concerns.

We look forward to receiving your revised manuscript. 

Sincerely,

Thomas McBride, PhD

Senior Editor 

PLOS Medicine

plosmedicine.org

1- Thank you for noting the restrictions on data sharing. Please edit the statement to read:

“The data used in this study are available on request to the ALSPAC Executive (alspac-exec@bristol.ac.uk). The ALSPAC data management plan describes in detail the policy regarding data sharing. Full instructions for applying for data access can be found

here: http://www.bristol.ac.uk/alspac/researchers/access/. The ALSPAC study website

contains details of all the data that are available (http://www.bristol.ac.uk/alspac/researchers/our-data/).”

2- Please revise your title according to PLOS Medicine's style. Please place the study design (e.g, "a cohort study”) in the subtitle (ie, after a colon).

3- In the Abstract Methods and Findings, Please include the study design, population and setting, number of participants and relevant demographics (e.g., mean age, sex, race/ethnicity), years during which the study took place, length of follow up, and main outcome measures.

4- In the Abstract, and throughout the manuscript (Results and Tables), please include p values alongside 95% CIs for comparisons.

5- In the Abstract Methods and Findings, please include the actual amounts and/or absolute risk(s) of relevant outcomes, not just relative risks or correlation coefficients. (example for absolute risks: PMID: 28399126). 

6- Abstract, line 31: please define the p-value thresholds used for these associations and include a bit more explanation.

7- In the last sentence of the Abstract Methods and Findings section, please describe the main limitation(s) of the study's methodology.

8- At this stage, we ask that you include a short, non-technical Author Summary of your research to make findings accessible to a wide audience that includes both scientists and non-scientists. The Author Summary should immediately follow the Abstract in your revised manuscript. This text is subject to editorial change and should be distinct from the scientific abstract. Please see our author guidelines for more information: https://journals.plos.org/plosmedicine/s/revising-your-manuscript#loc-author-summary

9- Did your study have a prospective protocol or analysis plan? Please state this (either way) early in the Methods section.

10- Please call out the ethics section in the Methods with its own subheading. Additionally, please specify if consent was written or oral.

11- Methods, Line 148: at what time?

12- Please present and organize the Discussion as follows: a short, clear summary of the article's findings; what the study adds to existing research and where and why the results may differ from previous research; strengths and limitations of the study; implications and next steps for research, clinical practice, and/or public policy; one-paragraph conclusion.

Comments from the reviewers:

Reviewer #1: The article by Khouja et al examines the the genetic contributions to smoking initiation with e-cigarette use in young adults. Overall, the article is well written and highlights the importance of genetics in mediating the outcome measures. 

i. The primary factor missing from the results is an overall understanding of the types of genetic factors that may contribute to the findings. Thus, a more comprehensive description of the makeup of the genes that contribute to the polygenic risk scores would be helpful. 

ii. Also, of the 378 single nucleotide polymorphisms that went into the smoking initiation polygenic risk scores, could the authors identify which of these genetic variants have previously been identified as functional? A brief discussion of the relevance of the outcome would be helpful for the reader/reviewer. 

iii. Further, could the authors comment on how important their findings are when OR are between OR = 1.24 - 1.29? How is this similar or different if only one genetic polymorphism was evaluated at the most significant p-value vs. smoking initiation PRS, for example?

Reviewer #2: This is an interesting study on the association of genetic liability to smoking initiation with e-cigarette use in young adults. However, there are a few major issues needing attention.

1) The authors showed the some associations between PRS and e-smoking, and also some associations between PRS and smoking. However, the 3-way relationship has not been systematically untangled. To account for the smoking impact between the PRS and e-smoking, for example, mediation analysis could be used while smoking is the mediator.

2) Not convinced why different P-value thresholds were used. The 5x10-8 one ended up with 347 SNPs should be used as it took account of proper multiple test adjustment. All the others are not adequate and irrelevant.

3) It said in a few places that the analyses were adjusted by principal components 1-10. But principal components of what? Never explained anywhere in the paper.

4) Still not clear and convinced what all these negative controls were for.

5) Description of the datasets being used and selected was difficult to follow and confusing.

6) Statistics analyses section is too brief and could be more comprehensive and detailed. By the way, what are the 'first 10 principal components'?

Reviewer #3: This manuscript describes the results of a study that examined the use of polygenic risk scores to predict ever use of e-cigarettes. A large cohort from multiple studies with smoking initiation, PRSs, and e-cigarette use assessed was used. The analyses included assessment of negative controls (e.g., taking risks) in order to examine biological causality. The analytic approach focused on predicting longitudinal ever e-cigarette use based on PRSs. Overall, this study addresses an under-studied area of research, the results broaden our understanding of the etiology of e-cigarette use, and the findings can support useful policy implications and subsequent analytic approaches. Addressing the relatively minor concerns below may strengthen this important study.

1. Limitations also include variability in the nature of key assessments and the use of self-report data, which should be mentioned.

2. More information about the potential policy implications of the results should be offered.

Reviewer #4: Please see review document attached.

[LINK]

---

## [Decision Letter · Decision Letter 2]

11 Jan 2021

Dear Dr. Khouja,

Thank you very much for re-submitting your manuscript "Association of genetic liability to smoking initiation with e-cigarette use in young adults; A cohort study." (PMEDICINE-D-20-02674R2) for review by PLOS Medicine.

I have discussed the paper with my colleagues and the academic editor and it was also seen again by two of the original reviewers. I am pleased to say that provided the remaining editorial and production issues are dealt with we are planning to accept the paper for publication in the journal.

[LINK]

We look forward to receiving the revised manuscript by Jan 18 2021 11:59PM.   

Sincerely,

Thomas McBride, PhD

Senior Editor 

PLOS Medicine

plosmedicine.org

Requests from Editors:

1- Please ensure that the study is reported according to the STROBE guideline, and include the completed STROBE checklist as Supporting Information. Please add the following statement, or similar, to the Methods: "This study is reported as per the Strengthening the Reporting of Observational Studies in Epidemiology (STROBE) guideline (S1 Checklist)."

2- You note that your analyses were pre-planned but I didn’t see an analysis plan in the SI files. If one exists (e.g., in an IRB or grant application), please include this in the SI files and reference it from the Methods section. If no such document exists, please note this in the Methods section.

3- Perhaps "Tobacco smoking ..." at line 17, and “tobacco smokers” at line 72?

4- Line 45, remove “Taken together”.

5- It seems that you use “e-ciggarette use” and “vaping” interchangeably; perhaps state "e-cigarette use or 'vaping'" early on? At line 65, do you mean "... both vaping and using cigarettes ..."?

6- Please begin the Discussion with a brief description of what was done before summarizing results.

7- In table 3, perhaps add more contest in the “Number of sexual partners by 23” outcome, to avoid misinterpretation of “n=2,505" one column over.

8- Please remove the funding and COI statement can go from the main text and make sure this information is captured in the appropriate sections of the submission form.

9- References 1 and 19 look incomplete; Ref 3 needs reformatting and the journal name; Ref 4, 28, should note that they’re preprints.

Comments from Reviewers:

Reviewer #2: Many thanks authors for their great effort to improve the manuscript. I am satisfied with the response and revision. No further issues needing attention.

Reviewer #4: The authors have adequately addressed the reviewer comments. I have no additional comments to provide.

[LINK]

---

## [Editor Report · Decision Letter 3]

31 Jan 2021

Dear Dr Khouja, 

On behalf of my colleagues and the Academic Editor, Dr Hall, I am pleased to inform you that we have agreed to publish your manuscript "Association of genetic liability to smoking initiation with e-cigarette use in young adults: A cohort study." (PMEDICINE-D-20-02674R3) in PLOS Medicine.

PRESS

Sincerely, 

Richard Turner, PhD 

rturner@plos.org